# Enhancement of Chondrogenic Markers by Exosomes Derived from Cultured Human Synovial Fluid-Derived Cells: A Comparative Analysis of 2D and 3D Conditions

**DOI:** 10.3390/biomedicines11123145

**Published:** 2023-11-25

**Authors:** Bo Han, William Fang, Zhi Yang, Yuntao Wang, Shuqing Zhao, Ba Xuan Hoang, C. Thomas Vangsness

**Affiliations:** 1Department of Surgery and Biomedical Engineering, Keck School of Medicine, University of Southern California, 1333 San Pablo St., BMT 302A, Los Angeles, CA 90089, USAbaxuanho@usc.edu (B.X.H.); 2Department of Orthopaedic Surgery, Keck School of Medicine, University of Southern California, Los Angeles, CA 90089, USA

**Keywords:** exosome, chondrogenesis, synovial fluid-derived cells, 3D encapsulation, cartilage repair

## Abstract

Objective: The goal of this pilot study was to investigate the effects of exosomes derived from synovial fluid-derived cells (SFDCs) cultured under normoxic conditions in a two-dimensional (2D) monolayer or encapsulated within a three-dimensional (3D) matrix for chondrogenic differentiation in vitro and cartilage defect repair in vivo. Design: Synovial fluid samples were obtained from three patients, and SFDCs were isolated and expanded either in a 2D monolayer culture or seeded within a transglutaminase cross-linked gelatin (Col-Tgel) to create a 3D gel culture. Exosomes derived from each environment were isolated and characterized. Then, their effects on cartilage-cell proliferation and chondrogenic differentiation were assessed using an in vitro organoid model, and their potential for enhancing cartilage repair was evaluated using a rat cartilage defect model. Results: SFDCs obtained from different donors reached a state of senescence after four passages in 2D culture. However, transferring these cells to a 3D culture environment mitigated the senescence and improved cell viability. The 3D-cultured exosomes exhibited enhanced potency in promoting chondrogenic differentiation, as evidenced by the increased expression of chondrogenic genes and greater deposition of cartilage-specific extracellular matrix. Furthermore, the 3D-cultured exosomes demonstrated superior effectiveness in enhancing cartilage repair and exhibited better healing properties compared to exosomes derived from a 2D culture. Conclusions: The optimized 3D culture provided a more favorable environment for the proliferation of human synovial cells and the secretion of exosomes compared to the 2D culture. The 3D-cultured exosomes exhibited greater potential for promoting chondrogenic gene expression in vitro and demonstrated improved healing properties in repairing cartilage defects compared to exosomes derived from the 2D culture.

## 1. Introduction

Articular cartilage degeneration and the associated arthritis represent a significant growing burden worldwide with the aging human population. Cartilage has a limited capacity for spontaneous healing due to poor blood supply and low metabolic activity [1]. Aging represents one of the most common risk factors, but other factors include gender, obesity, genetics, diet, and even mechanical stress [2]. Osteoarthritis (OA) is a chronic degenerative disease that occurs when the cartilage in the joints gradually wears down, causing swelling, stiffness, and pain. The condition can affect any joint but most commonly occurs in the hands, hips, spine, and knees [3].

Inside the knee joint cavity, synovial fluid surrounded by the synovial membrane protects and cushions the articular cartilage The synovial membrane is composed of two layers: the intima or surface layer of cells consists of macrophages and fibroblasts, and the subintima, or underlying tissue, which is composed of synoviocytes, blood, and lymphatic vessels [4]. Synovial fluid-derived mesenchymal stem/stromal cells (SF-MSCs) are fibroblast-like cells isolated from the synovial fluid with a phenotype similar to bone marrow MSC (BM-MSC). Mesenchymal stem cells or stromal cells have potent anti-inflammatory/immunosuppressive properties, which underlie much of their therapeutic potential [5,6]. Studies have shown that the number of SF-MSCs increases by approximately 100-fold in injury or osteoarthritis cases [7], suggesting that the SF-MSCs may participate directly or recruit other cells to participate in the body’s healing response [8]. Characterization studies carried out on SF-MSCs demonstrated higher levels of chondrogenesis and osteogenesis compared to MSCs derived from bone marrow or dental pulp [9]. While pre-clinical animal studies have shown the effectiveness of intra-articular injections of SF-MSCs ameliorating cartilage defects in rat knees [10] and the ability for chondrocyte regeneration and maturation in a rabbit model [11].

In this study, synovial fluid-derived cells (SFDC) were chosen due to their accessibility, enhanced chondrogenic potential, and demonstrated resistance to IL-1β-induced stress, as outlined in a previous manuscript [12]. However, cellular therapies have many disadvantages, including the fact that MSC numbers and phenotypes depend highly on the donors’ characteristics [13]. While there are also possible clinical safety issues, including that of transplanted allogeneic MSCs not being immune-privileged [14], the generation of antibodies and immune rejection have been previously reported [15,16].

Extracellular vesicles or exosomes avoid these issues while still demonstrating the therapeutic properties of the host cell [17]. Exosomes are nano-sized (40–150 nm) extracellular vesicles originating from the internal budding of the plasma membrane during endocytic internalization [18,19]. They are released by most cell types and are considered part of the intercellular communication system, carrying cargo including proteins, messenger and microRNAs, lipids, and metabolites that reflect the biological state of the parent cells [20,21]. They act as messengers and transfer these small molecules to other cells, acting in a paracrine or even an endocrine manner to modify the behavior of adjacent or distant cells [22]. Experimental evidence has shown that exosomes derived from SFDCs (SFDC-Exos) could reduce OA symptoms in a rat model [23], together with other evidence that exosomes derived from embryonic stem cells helped to heal osteochondral defects in adult rats [24] and intra-articular injections of stem cell-derived exosomes alleviated cartilage destruction and matrix degradation in a destabilized medial meniscus (DMM) rat model [25].

Developing exosome technology for therapeutic applications is challenging due to the requirement for large quantities of exosomes. Thus, the isolation method must be scalable and preserve the clinical characteristics during large-scale manufacturing [26]. Typically, exosomes are isolated from a culture medium in which cells are expanded in a monolayer culture [27]. To improve exosome efficacy and yield, 3D cultures and hollow fiber bioreactors have recently been proposed [28,29,30,31]. Additionally, studies have shown that the composition of exosomes may vary depending on the culture conditions used to produce them. For example, exosomes derived from cells cultured in different media or under varying oxygen levels differ in terms of protein and RNA content and biological functions [32,33]. Similarly, the use of different culture surfaces or 3D-culture systems affects the size, quantity, and cargo of exosomes released by cells [34]. Despite these findings, there is a lack of studies characterizing the effects of using 2D culture versus 3D culture for harvesting and isolating exosomes from SFDCs and testing their biological properties. This study aims to compare the exosome yield and therapeutic effects of SFDC-Exos harvested from 2D and 3D cultures.

## 2. Materials and Methods

### 2.1. Study Subjects

All methods were performed in accordance with the relevant guidelines and regulations of the University of Southern California and were approved by its IRB committee. The synovial fluid samples were obtained from a study registered at ClinicalTrials.gov (NCT03242707), and all subjects were recruited at USC Keck Hospital. The patients all provided written consent before the study. These patients (Figure 1A) all had mild to moderate osteoarthritis and were randomized into two groups, one receiving an intra-articular injection of hyaluronic acid (HA) or a harvested and processed adipose tissue. During clinic visits, a sample of their synovial fluid was taken and analyzed before the injection, at baseline, after 6 weeks, 3 months, and 6 months follow-up. The synovial fluid sample used in this study was taken at baseline before any injections. 

### 2.2. Sample Processing

The synovial fluid samples were collected through an intra-articular joint ultrasound aspiration and injected into EDTA-containing tubes. Fluid samples were diluted 1:1 with 1× DPBS solution and centrifuged to isolate cells at 400× *g* for 7 min. The supernatant was stored at −80 °C for subsequent testing, and cells were processed for culture. 

### 2.3. Culturing Human Synovial Fluid-Derived Cells (SFDC)

The cells were allowed to attach and expand in T25 flasks in media (DMEM +10% FBS + 1% PS) at 5% CO_2_ and incubated at 37 °C. Media was changed every 3–4 days. When the cells reached 70% confluence, the cells were passaged by trypsinization or harvested for assays. SFDC were seeded in a 6-well plate for the monolayer culture for 6 days before marker staining or RNA extraction. 

In the 3D culture, the cells were embedded in 6% gelatin crosslinked with transglutaminase [35]. Briefly, 10^5^ SFDC cells were aliquoted into 1 mL of 6% gel and suspended before 50 μL of transglutaminase was added. Then, 20 μL of the cell-gel mixture was cast on the surface of each well of a 48-well suspension cell culture plate. The plates were incubated at 37 °C for 1 h to solidify the gel, followed by addition of 500 μL of culture medium to submerge a half-dome-shaped 3D construct. Medium was changed every 2–3 days. 

### 2.4. Cell Proliferation Assay

Cell counting was performed to determine the cell growth and doubling times. In the 2D culture, the monolayer cells were harvested using 0.05% trypsin-EDTA 6 days after seeding. In the 3D culture, the doubling times were deduced from two-time points (6 and 12 days after seeding). The Col-Tgel cell constructs were washed twice with PBS and digested with 0.05% trypsin-EDTA solution for 4 h to release cells. All the cell samples were collected and counted with a cell counter (Beckman Coulter, Brea, CA, USA).

### 2.5. Isolation of Exosomes

Exosomes were isolated from the cell culture medium using a precipitation-column protocol developed by our lab after comparing different protocols for their yields in pilot studies. The isolation and harvesting technique can be found in Figure 2A. Cells used for the exosome isolation in either 2D or 3D were passage 3 cells. Cells were cultured in a complete medium for 6 days and changed into an exosome-free, serum-free, conditioned medium for 48 h. The culture medium was harvested and centrifuged at 2000× *g* for 10 min to remove cell debris and larger vesicles. Then PEG8000 solution was added to make a final PEG concentration of 12% to precipitate exosomes at 4 °C overnight. The solution was then centrifuged to pellet the vesicles. Exosome pellets were resuspended with 1 mL of 1× PBS and purified by ultrafiltration using a qEV column (iZON Science, Christchurch, New Zealand) following the manufacturer’s suggested protocol. The exosomes harvested from fraction collection were then characterized to ensure that they met the definition of an exosome [36].

### 2.6. Exosome Characterization

The number and size of the isolated SFDC-derived exosomes were assessed by nanoparticle tracking analysis (NTA) by using a NanoSight NS300 (Malvern Panalytical Ltd, Malvern UK) equipped with an sCMOS camera (Oxford Instruments, Abingdon, UK). The prepared exosome sample was diluted 200- or 400-fold with 1× PBS buffer to reach the optimal concentration for instrument linearity. Readings were taken at a camera level set to 13 with manual temperature monitoring. The camera captured a video file of the particles moving under Brownian motion and was used to measure the particle size and concentration. The software tracks the number of particles and calculates their hydrodynamic diameters. Data were analyzed with the NTA software version 3.1 (build 3.1.54) (Malvern Panalytical Ltd., Malvern, UK) with a detection threshold set to 5 and blur and Max Jump Distance set at auto. 

Exosomal protein markers CD9, CD63, and CD81 were determined by using the Exosome ELISA Complete Kit (System Biosciences #EXOEL-CD9A-1, #EXOEL-CD63A-1, and #EXOEL-CD81A-1). The exosome solution was processed according to the manufacturer’s instructions. Briefly, protein standards and exosome protein samples were placed in the microtiter plate wells overnight at 37 °C. After washing with the working buffer, 50 µL of anti-CD63, anti-CD9, or anti-CD81 (1:100) was added to the wells and incubated for 1 h on a shaker at 37 °C. After washing 3 additional times, a secondary antibody (1:5000) was added for 1 h at room temperature. A spectrophotometric plate reader performed colorimetric (super-sensitive tetramethylbenzidine) determination at OD450 nm (Molecular Devices, San Jose, CA, USA). The exosomal protein marker concentration was derived from the standard curve. Tetraspanin CD63 protein concentration was used as a representative exosomal marker protein in the current study [37]. 

### 2.7. Exosome on Chondrocyte Viability

To test the effect of exosomes on cartilage cell viability, a 2D assay was performed. Briefly, P4-P5 human chondrocyte (HC, VWR International, Randor, PA, USA) cells were seeded at a density of 5 × 10^3^ cells/well in a 96-well plate and allowed to attach overnight. Exosomes were added to the wells at a concentration of 25 μg/mL for 48 h. A 2D cell viability assay was performed using MTT staining during the final 4 h of incubation. After washing the cells with PBS, they were treated with MTT working solution (0.5 mg/mL MTT in culture medium) for 4 h. The resulting formazan crystals were dissolved in DMSO and measured using a multi-plate reader (Molecular Devices, San Jose, CA, USA).

### 2.8. Exosomes on Chondrogenic Differentiation 

Human chondrocytes (HC) were expanded in DMEM/F-12 (Mediatech, Manassas, VA, USA) with 10% (*v*/*v*) fetal bovine serum (FBS) and 1% (*v*/*v*) penicillin-streptomycin (PS) in a humidified atmosphere of 95% O_2_/5% CO_2_ at 37 °C. The cells were subcultured using 0.25% trypsin/EDTA when they reached 70% confluence. Passage 5 HC cells were collected to investigate the effect of exosomes on chondrogenic functional studies using a 3D chondrogenic differentiation model. A graphical illustration of this model can be found in the Results section. To create the Col-Tgel cell mixtures, 100 µL of a 6% gelatin solution (Sigma-Aldrich, Saint Louis, MO, USA) was mixed with 2 × 10^5^ HCs and 5 µL of transglutaminase. Aliquots (20 μL) of the mixture were seeded into 48-well suspension plates, and once the gel had solidified, 0.5 mL of growth medium was added with 25µg/mL supplementing either 2D- or 3D-derived exosomes. The medium was changed every 3–4 days to study the chondrogenic differentiation. On day 14, samples were collected and analyzed for proteoglycan deposition using toluidine blue (TB) staining and colorimetric assays [38]. The constructs were fixed with paraformaldehyde and stained with TB overnight; images were then captured to analyze the unique characteristics of HCs. For the TB colorimetric assay, the constructs were incubated overnight in 6 M guanidine hydrochloride, and the supernatant was collected for absorbance measurement using a microplate reader (Molecular Devices, Sunnyvale, CA, USA). 

The embedded cell construct was also fixed in 10% neutral formalin for type II collagen staining. The construct was first treated with peroxidase suppressor solution (Thermo Scientific, Waltham, MA, USA) for 30 min, followed by incubation in the blocking buffer (5% BSA in TBST) for 30 min, and the sample was then incubated in the primary anti-Col 2A1 (Santa Cruz Biotech, TX, USA) overnight. The color was developed by a biotinylated secondary antibody (1:800, Sigma, MO, USA) for 1 h. The color reaction was revealed by incubating the sample in a DAB working solution. 

### 2.9. Real-Time PCR

Total RNA was isolated from embedded cells on day 10 using TRIzol (Invitroge, Carlsbad, CA, USA) according to the manufacturer’s instruction to study the chondrogenic gene expressions. Briefly, total RNA (200–500 ng) was reverse-transcribed using a High-Capacity cDNA Reverse Transcription Kit (Applied Biosystems, Foster City, CA, USA). Real-time polymerase chain reactions (PCR) were performed using FastStart Universal SYBR Green Master Mix (Roche Diagnostics, Basel, Switzerland) in a LightCycler 96 (Roche Diagnostics, Basel, Switzerland). Quantification was performed relative to the levels of the housekeeping gene GAPDH. The data analysis was performed using the 2^−ΔΔCT^ method. Quantitative data are presented as means ± standard deviations of three independent PCR experiments.

### 2.10. Animal Studies

Fifteen 16-week-old male Sprague Dawley rats were utilized, and the animal experiments were reviewed and approved by the Institutional Animal Care and Use Committee (IACUC) of the University of Southern California. The study design included three groups (A, B, C, *n* = 5) to assess two different exosomes and their carrier control. Group A was the control group with only carrier Col-Tgel, while groups B and C were treated with 2D and 3D exosomes plus Col-Tgel, respectively. Anesthesia was induced through intraperitoneal injection of ketamine/xylazine. After aseptic preparation with a 10% betadine solution, both knee joints of each rat were sterilely draped, and an anteromedial approach was used to open the knee capsule. The patellae were laterally dislocated, and full-thickness articular cartilage defects (2 mm in diameter) were created in trochlear grooves by carefully drilling in a vertical direction using a 2 mm drill. Drilling was performed 3 mm deep through the subchondral bone. After removing cartilage and bone debris, the gel mixture with cells or exosomes was transplanted into the full-thickness defect in the experimental knee using a pipette. Animals were euthanized after 4 weeks, and joints were harvested for histology. H&E and safranin-O staining were performed, and the degree of cartilage repair was grossly assessed and scored using a modified O’Driscoll histological scoring system by two blinded personnel [39]. In brief, the system assigns scores based on four categories: structure, cellularity, matrix staining, and integration. Each category is evaluated on a scale of 0 to 4, with 0 being the worst and 4 being the best. The scores are then added together to obtain a total score ranging from 0 to 16.

### 2.11. Statistical Analysis

Numerical data are presented as ANOVA analyzed with the mean ± standard deviation (SD) and statistical difference. Statistical analysis was performed on SAS (SAS Institute, NC, USA). Values of *p* < 0.05 were considered to indicate statistically significant differences as follows * *p* < 0.05 and ** *p* < 0.01.

## 3. Results

### 3.1. Patient and Synovial Fluid-Derived Cell Characteristics

The study utilized synovial fluid aspiration from three patients as samples, and their demographics are presented in Figure 1A. The mean age of the patients was 65.3 (95% CI: 55.3–75.3), and the mean body mass index (BMI) was 26.4 (95% CI: 23.6–29.1). Two patients had K-L grade 2 knees, while one had a K-L grade 3 knee. The SFDC cells were grown as a monolayer (2D) on a regular plastic surface, and interestingly, the doubling times differed between the patients. Patient Y’s cells had the fastest growth, while patient Z’s had the slowest. However, after the fourth passage in the 2D monolayer culture, all three cell types showed reduced proliferation rates and entered senescence, as shown in Figure 1B. The P4 cells were transferred from the monolayer culture to a 3D hydrogel culture to study the environmental impact on cell morphology and viability. In Figure 1C, SFDCs grown in 2D had an elongated, fibroblast-like appearance, while those embedded in 3D displayed a spherical shape. Cell viability was also significantly increased (*p* < 0.01) in the 3D embedding, as exhibited in Figure 1D. 

### 3.2. Isolation and Characterization of SFDC Exosomes

The harvesting and isolation protocol for the exosomes is presented in Figure 2A, which allowed for quick and scalable exosome isolation without additional equipment. CD63, an exosome transmembrane marker, was used to quantify the exosomes in each collected fraction to confirm their presence. As shown in Figure 2B, fractions 9–11 showed a significant peak, were verified positive for CD63, and were thus chosen for further analysis. However, fractions 17–19, which also tested positive for CD63, were excluded from the analysis as they were extracellular vesicles (EVs) rather than exosomes. As the study was focused on exosome analysis, only EVs within the size range of exosomes were collected [40]. In addition, Figure 2C shows the results of Nanosight NTA analysis, which confirmed that the isolated exosomes were approximately 100 nm in size and had a round or oval shape.

### 3.3. Quantitative and Qualitative Comparison of Exosomes Derived from 2D and 3D Cell Cultures

Exosome markers CD9, CD63, and CD81 were quantified in exosomes collected from 2D and 3D cultures of P3 SFDCs from three patients using ELISA. As presented in Figure 3, exosome marker expression was normalized to cell numbers. Patient Y, with the fastest proliferating SFDCs, exhibited the highest exosome marker expression, while patient Z, with the slowest proliferating SFDCs, displayed the lowest exosome marker expression. The expression of the tetraspanin CD63 was found to be significantly higher than CD9 and CD81 (*p* < 0.05). Surprisingly, both the 2D- and 3D-culture conditions demonstrated a similar pattern of exosome surface marker expression, although there were some variations in relative expression. In the case of patient Y (PY), exosome markers were significantly more expressed in 3D cultures compared to 2D cultures. All three markers, CD63, CD81, and CD9, exhibited higher expression in the 3D culture (*p* < 0.01). For patient PX, CD63 and CD9 showed significantly higher expression in 3D culture compared to 2D (*p* < 0.05). In the case of patient Z, CD81 expression was notably higher in the 3D culture. In general, the trend indicated that the expression of exosome markers was higher in the 3D culture compared to the 2D culture. However, it is important to note that this observation is also specific to each individual patient.

### 3.4. The In Vitro Effects of SFDC Exosomes on Viability and Redifferentiation of Dedifferentiated Human Chondrocytes

To investigate the impact of SFDC exosomes on HC cell viability and morphology, human chondrocytes (HCs, P5) were supplemented with exosomes derived from P3 2D and 3D cultures from patient Y, while regular growth medium served as a control. After 48 h, cell viability and HC morphology showed no significant differences (*p* > 0.05) in cell viability among the three conditions, and HC morphology and cell density remained similar (Figure 4A,B).

The functional model for testing the effect of SFDC exosomes on chondrogenic differentiation involved seeding dedifferentiated HCs in a 3D matrix (Figure 4C). The HC cells were supplemented with exosomes from 2D or 3D SFDC cultures. After 14 days, the HC cells were stained with toluidine blue and anti-Col2A1 antibody to visualize proteoglycan and type II collagen deposition, indicators of chondrogenic potential (Figure 4D). The analysis showed a significant increase (* *p* < 0.05) in the number of positively stained cells and proteoglycan deposition across all three donors when treated with 3D exosomes compared to 2D exosomes, with the control group showing the lowest number of positively stained cells (Figure 4E). Real-time PCR analysis revealed significant upregulation of SOX9 (*p* < 0.01) and aggrecan (*p* < 0.01) mRNA expression levels after treatment with exosomes from 3D exosomes compared to 2D exosomes in all three donors. Conversely, the gene for collagen type I (Col1) was significantly downregulated, with a decrease in mRNA expression when treated with 3D exosomes compared to 2D exosomes in patients Y and Z (*p* < 0.05) (Figure 4F). The study demonstrated that exosomes derived from 3D culture conditions upregulated gene expression for chondrocyte differentiation in all three donors.

### 3.5. The In Vivo Effects of SFDC Exosomes on Cartilage Repair 

To investigate the potential of exosomes in cartilage regeneration, a rat cartilage defect model was utilized. After four weeks of implanting exosome-containing gel plugs, the rat knees were collected for macroscopic and histological analysis, as depicted in Figure 5A. H&E staining revealed the presence of new cartilage formation at the defect sites in all experimental groups. However, the group treated with Col-Tgel-containing P3 exosomes derived from patient Y displayed superior structural integration at the defect sites. Furthermore, both the 2D and 3D exosomes exhibited a more pronounced reparative effect compared to the vehicle-only group, as indicated by Safranin O staining (Figure 5B), demonstrating enhanced proteoglycan deposition. The evaluation of cartilage repair using the modified O’Driscoll histological scoring method (depicted in Figure 5C) revealed that both 3D-derived SFDC exosomes significantly outperformed the 2D exosomes (*p* < 0.05) and the Col-Tgel vehicle (*p* < 0.01) in terms of cartilage repair effect. These results indicate that exosomes derived from 3D-culture conditions possess therapeutic advantages over 2D-cultured exosomes when it comes to cartilage repair. 

## 4. Discussion

In this pilot study, we aimed to explore the potential of exosomes derived from synovial fluid-derived cells (SFDCs) in modulating human cartilage cells. We first isolated and characterized SFDCs from human synovium, and then investigated the influence of exosomes derived from these SFDCs on human cartilage cells in vitro. Subsequently, we evaluated the effects of these exosomes in an in vivo rat cartilage defect model. 

Osteoarthritis (OA) is characterized by the degeneration of cartilage, which presents a significant challenge for intrinsic repair due to its limited vascularity and regenerative capacity [41]. Previous studies have shown that SFDCs can contribute to cartilage repair when delivered in combination with a carrier or as a standalone injection [6]. For instance, when human SFDCs encapsulated in a hyaluronic acid gel carrier were utilized in a rat cartilage defect model, researchers observed the formation of smooth cartilage with a significant reduction in the defect area [10]. Similarly, in a porcine study, SFDCs embedded in a platelet-rich plasma (PRP) hydrogel led to increased cell growth and chondrocyte proliferation [42]. In a rabbit model, direct intra-articular injection of SFDCs resulted in improved cartilage healing outcomes [11]. Our study aligns with these findings, as we observed similar positive effects on chondrogenesis through the evaluation of biomarkers and functional studies.

In our study, we encountered challenges related to the expansion of SFDCs for exosome production. Typically, cell expansion can lead to senescence, causing cells to stop proliferating. To overcome this limitation, we employed a 3D matrix to encapsulate SFDCs, which allowed us to overcome senescence and regain their proliferation potential. This 3D-culture condition also mimicked the hypoxic environment of SFDCs in their native in vivo conditions in the synovial space or cartilage, as opposed to exposure to 21% oxygen in typical 2D culture. Our findings suggest that hypoxia and mechanical loading enhance proliferation potential and promote chondrogenic differentiation ability. Previous studies have also shown that mechanical stimulation and construct stiffness in the cellular environment can affect chondrogenesis and articular cartilage tissue regeneration in vitro [43,44,45]. Additionally, hypoxia can alter the expression of extracellular vesicle (EV) contents and impact their biological functions [46]. 

To harvest and isolate exosomes, we developed a protocol combining polyethylene glycol (PEG) precipitation with ultrafiltration to maximize yield and quality. This method offers scalability advantages without volume limitations or the need for specialized instrumentation. However, a potential drawback of using PEG precipitation for EV isolation is that the polymer can precipitate other protein components from the culture medium. To address this issue, we employed additional purification techniques, including size exclusion chromatography, and selectively collected fractions containing the tetraspanin marker CD63. While ultracentrifugation is a commonly used approach for exosome isolation, it has several drawbacks such as being labor-intensive, time-consuming, requiring a large amount of starting material, and yielding low exosome quantities [47]. 

In terms of exosome morphology, there were no discernible differences between exosomes obtained from 2D or 3D cultures. Both sets of exosomes exhibited a spherical shape, were around 100 nm in size, and contained tetraspanin markers CD9, CD63, and CD81. Interestingly, the expression of exosome markers varied between patients, with patient Y showing faster SFDC doubling times and higher exosome marker expression, while patient Z had slower doubling times and lower exosome marker expression. Previous research has linked exosome concentration to the expression of the CD63 marker [37]. Therefore, our study suggests that SFDCs cultured in a 3D environment produced more exosomes based on marker quantification. However, it is important to note that tetraspanins are widely distributed in the plasma membrane and may be present in other types of vesicles [48,49]. Some studies have shown that markers CD63 and CD81 are enriched in vesicles that bud from the plasma membrane, while other studies have demonstrated the secretion of exosomes lacking CD63 [50,51]. Further studies with larger sample sizes are needed to determine whether these markers correspond to differences in parent cell proliferation capability or affect exosome yield.

The study investigated whether exosomes produced by synovial fluid-derived cells were involved in modulating HC proliferation and differentiation. In vitro testing of the culturing of HCs with exosomes harvested from 2D and 3D conditions showed no significant difference in influencing chondrocyte proliferation or morphology compared to the control. Cytotoxicity studies from a CCK-8 analysis also showed no significant differences in the cell viability of chondrocytes treated with exosomes. Exosomes from immune cells have demonstrated cytotoxicity effects on tumors and cancer cells [52]. However, the effect of synovial fluid exosomes on cells is lacking in the literature. These exosomes have been shown to be potential biomarkers for different stages of osteoarthritis and inflammation [53].

In vitro functional tests are important for determining exosome properties and function. The model used in this study used dedifferentiated human chondrocytes (P5) seeded in a 3D matrix to test the exosome effect on chondrogenic differentiation. In general, chondrocytes are considered dedifferentiated after an average of five passages [54]. Dedifferentiated chondrocytes gradually lose molecular markers that define differentiated chondrocytes, including the expression of markers of chondrogenesis and the expression of ECM molecules: GAGs, ACAN, SOX-9, aggrecan, and Col1 [55,56]. Typically, investigators supplement culture medium with different growth factors, change culture conditions, or cell density to slow the dedifferentiation process [46]. The current study used exosomes (P3) harvested from 3D and 2D cultures to test their ability to activate these dedifferentiated chondrocytes and promote the differentiation along the chondrogenic pathway. 

Lastly, there were significant differences in proteoglycan deposition in the rat cartilage defect model [57]. The 3D-cultured exosomes demonstrated a significantly greater number of stained colonies compared to 2D and control ones. The 3D-cultured exosomes also had significantly higher cartilage repair scores than both the 2D-cultured exosomes and vehicle. A possible mechanism behind the exosome’s effects remains to be determined but is hypothesized to mediate inflammation and release multiple transcription factors to promote cartilage regeneration [58,59,60]. 

## 5. Limitations

The present study had a limited sample size, with only three samples obtained from patients with osteoarthritis (OA). Given the variations in protein and cytokine expression among individuals [61,62], it is important to acknowledge that these differences could influence the quantity and quality of exosomes. To address this issue of donor variability, a broader range of donors, including both healthy individuals and patients with confirmed knee OA, should be included in future studies. Additionally, further studies in silico could be carried out. 

In this study, tetraspanin CD63 protein concentration was utilized as a representative marker for exosomes. However, it is worth noting that CD63 is a protein associated with the cell surface and is typically found in compartments of intracellular endosome/lysosomal origin. Consequently, CD63 may be present not only in exosomes but also in non-exosomal fractions of different densities. Therefore, it is essential to incorporate additional markers to validate the accuracy of using CD63 as a marker to distinguish exosomes from non-exosomal populations.

Furthermore, to establish the therapeutic potential of exosomes in the context of osteoarthritis, it is imperative to conduct larger-scale animal studies using relevant OA models. These studies will provide more comprehensive insights into the efficacy and effectiveness of exosome-based therapies in treating osteoarthritis.

## 6. Conclusions

The utilization of a 3D gel culture was shown to create a more favorable environment for the proliferation of synovial fluid-derived cells (SFDC) and the secretion of exosomes, in comparison to 2D culture. Exosomes obtained from the 3D-culture system exhibited enhanced chondrogenic differentiation in functional studies, indicating a greater capacity for promoting the repair and regeneration of chondrogenic tissue compared to exosomes derived from 2D culture. Moreover, these exosomes derived from 3D culture demonstrated superior cartilage regeneration capabilities in in vivo rat models. In conclusion, these findings present promising evidence supporting the potential therapeutic application of exosomes in cartilage regeneration. However, further investigations are necessary to elucidate the underlying mechanisms involved and to optimize the clinical use of exosomes in this context.

## Figures and Tables

**Figure 1 biomedicines-11-03145-f001:**
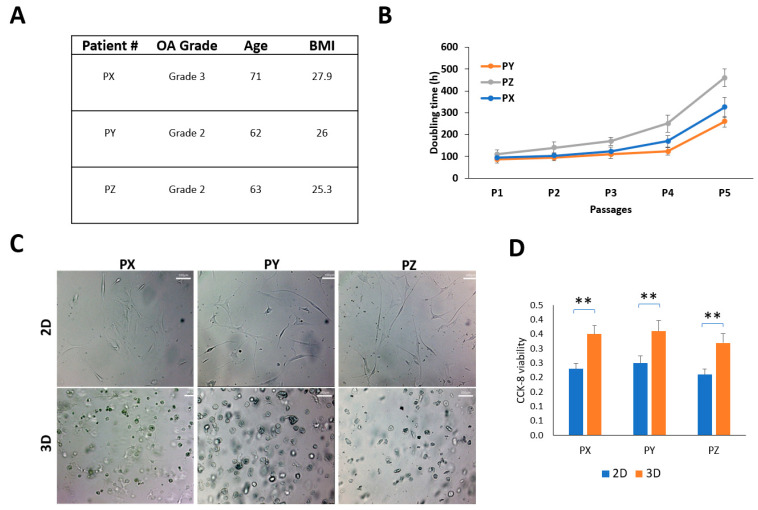
Patient-derived synovial cells and characterization. (**A**) Patient demographics; (**B**) harvested SFDC cultured on a 2D plate and their doubling times (hours); (**C**) SFDC morphology in 2D (P4) and in 3D (P5), scale bar = 100 µm. (**D**) Cell viability of SFDC in 2D (P4) and 3D (P5) by CCK-8 assay, ** *p* < 0.01.

**Figure 2 biomedicines-11-03145-f002:**
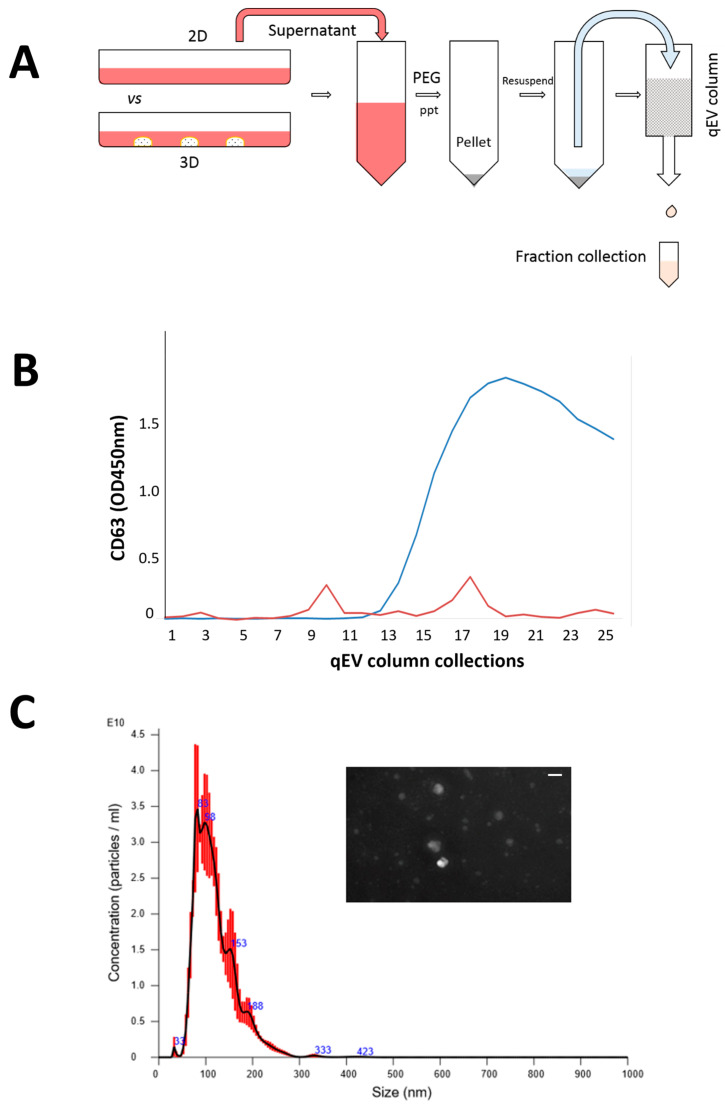
Isolation of SFDCC exosomes and characterization; (**A**) two-step protocol to purify exosomes by precipitation and ultrafiltration; (**B**) confirmation of purified fraction by exosome marker CD63; (**C**) vesicle size distribution validated by NanoSight NTA instrument; insert is Nanosight image of exosomes isolated from SFDC, scale bar = 100 nm.

**Figure 3 biomedicines-11-03145-f003:**
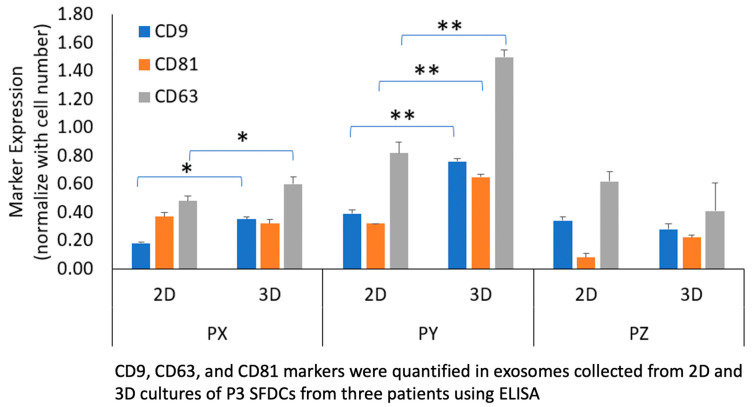
Comparison of exosome tetraspanin markers CD9, CD63, CD81 between culture conditions and patient population. * *p* < 0.05, ** *p* < 0.01.

**Figure 4 biomedicines-11-03145-f004:**
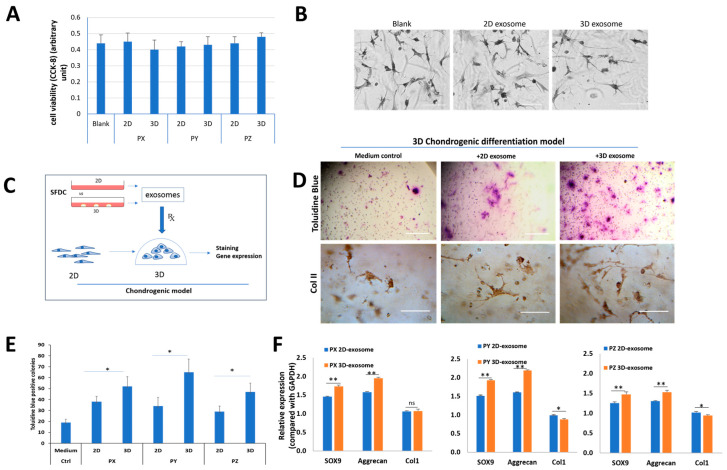
The effects of SFDC exosomes on human chondrocytes. (**A**) Cell viability of HCs supplemented with exosomes for 48 h, tested by CCK-8 assay; (**B**) the effect of SFDC exosomes derived from 2D and 3D culture on in vitro chondrocyte differentiation; (**C**) graphical representation of SFDC chondrogenic model testing. Exosomes were harvested from 2D- and 3D-cultured SFDCs. Exosomes supplemented the 3D culture of dedifferentiated human chondrocytes. Re-differentiation human chondrocytes were validated by in situ chondrogenic-marker staining and gene expression; (**D**) toluidine blue and type II collagen staining for the difference in proteoglycan deposition and ColII supplemented with control medium, 2D and 3D exosomes, scale bar = 500 µm for toluidine blue, scale bar = 100 µm for ColII IHC; (**E**) quantitated toluidine blue positive colonies treated with exosomes. * *p* < 0.05; (**F**) gene analysis for chondrogenesis of chondrocytes stimulated by different exosomes. GAPDH was used as a loading control, * *p* < 0.05, ** *p* < 0.01.

**Figure 5 biomedicines-11-03145-f005:**
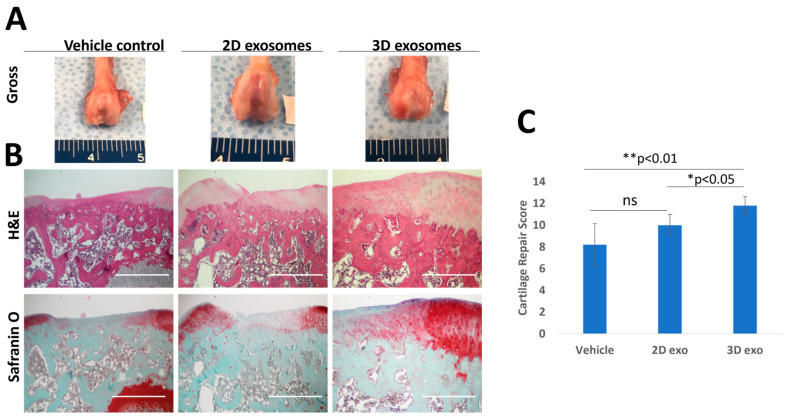
The effects of SFDC exosomes on cartilage defect repair in rats. (**A**) Gross morphology of rat knees four weeks post-exosome treatment (scale unit: mm). (**B**) Representative H&E and Safranin O staining at the defect sites. Scale Bar = 1000 µm. (**C**) Modified O’Driscoll histological scores, * *p* < 0.05, ** *p* < 0.01.

## Data Availability

The authors confirm that the data supporting the findings of this study are available within the article.

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
