# Peer review of "Enhancement of Chondrogenic Markers by Exosomes Derived from Cultured Human Synovial Fluid-Derived Cells: A Comparative Analysis of 2D and 3D Conditions"

_biomedicines, 2023, doi:10.3390/biomedicines11123145_

Round 1

Reviewer 1 Report

Comments and Suggestions for Authors

Overall Review:

The study aimed to the effects of exosomes derived from cultured human synovial fluid-derived cells on chondrogenic differentiation in vitro and cartilage defect repair in vivo. The study investigates the potential of exosomes to enhance chondrogenic markers and improve cartilage repair. The authors conducted experiments and analyzed the results to determine the effectiveness of exosomes in promoting chondrogenic differentiation and cartilage repair. The article provides insights into the potential of exosomes as a therapeutic approach for cartilage-related injuries and diseases.

Paper Weakness:

(1)   The text mentions that senescence began in P4 cells. What generation of cells were the extracted exosomes from? Please explain.

(2)   The image in Figure 2C is blurry. Please replace it with a high-resolution image.

(3)   In Figure 4D, which patient's exosomes were used, and which generation were they from? Please specify.

(4)   The author emphasizes the relevance of exosome quality in relation to their source (different patients) and the culture conditions (2D or 3D). So, which of these factors plays a primary role?

(5)   In Figure 4, Western blot (WB) analysis can be employed to bolster the experimental results.

(6)   In Figure 5, there is a similar issue. In investigating the impact of SFDC exosomes on rat cartilage defect repair, which generation and from which patient were the exosomes used?

(7)   There are issues with the experimental group setup in Figure 5, and it is necessary to include a control group without modeling. Furthermore, there is limited validation through experiments, and uCT as well as immunohistochemical staining should be used to assess the repair effect.

(8)   Most recent studies such as doi: 10.12336/biomatertransl.2023.02.005; doi: 10.12336/biomatertransl.2023.01.004, etc., are recommended to be cited in proper places.

(9)   The bibliography is deficient in discussion. There have been numerous other publications about bone fracture and bone loss (Sci Adv. 2023;9(14):eabo7868. doi:10.1126/sciadv.abo7868; Bioact Mater. 2022;23:156-169. Published 2022 Nov 11. doi:10.1016/j.bioactmat.2022.10.028; Cell Rep Med. 2023;4(1):100881. doi:10.1016/j.xcrm.2022.100881)

Comments on the Quality of English Language

Minor editing of English language required

Reviewer 2 Report

Comments and Suggestions for Authors

The article “Enhancement of Chondrogenic Markers by Exosome Derived from Cultured Human Synovial Fluid Derived Cells: A Comparative Analysis of 2D and 3D Conditions” by Han et al. focuses to compare the exosome yield and therapeutic effects of pare the exosome yield and therapeutic effects of SFDC-Exosomes harvested from 2D and 3D cultures-Exosomes harvested from 2D and 3D cultures. The research work is presented with proper methodology and results, discussed with appropriate way meeting the aim of work. Minor revision suggested to improve the article as follows.

The comments are as follows:

1.       Figure 2 C, image of exosomes isolated from SFDC need to high resolution.

2.       Discussion: in vivo rat cartilage defect model, invivo should be italicised. Add citation of rat as a suitable model.

Author Response

Thank you to the reviewer,

Round 2

Reviewer 1 Report

Comments and Suggestions for Authors

Experimental grouping and picture quality (such as scale bar) of the article need to be further improved.